# Body Image, Social Physique Anxiety Levels and Self-Esteem among Adults Participating in Physical Activity Programs

**DOI:** 10.3390/diseases11020066

**Published:** 2023-04-27

**Authors:** Afroditi Zartaloudi, Dimitrios Christopoulos, Martha Kelesi, Ourania Govina, Marianna Mantzorou, Theodoula Adamakidou, Loukia Karvouni, Ioannis Koutelekos, Eleni Evangelou, Georgia Fasoi, Eugenia Vlachou

**Affiliations:** 1Department of Nursing, Faculty of Health Sciences, University of West Attica, 12243 Athens, Greeceevlachou@uniwa.gr (E.V.); 2Psychiatric Hospital of Attica, 12462 Athens, Greece

**Keywords:** social physique anxiety, body image, BMI, self-esteem, physical activity, athletics

## Abstract

(1) Background: individuals may benefit from being involved in physical and athletic activities in order to improve their body appearance and promote their physical and mental health. This study aimed to investigate body image, body mass index (BMI) characteristics, social physique anxiety, self-esteem and possible correlations between the above factors. (2) Methods: 245 adults engaged in training programs in gyms, as well as in track and field, football and basketball athletic activities completed (a) a sociodemographic questionnaire which recorded their BMI values and utilized the (b) Body-Esteem Scale for Adolescents and Adults, (c) the Social Physique Anxiety Scale and (d) the Rosenberg Self-Esteem Scale. (3) Results: Females and individuals with higher BMI reported statistically significant lower body-esteem and greater social physique anxiety levels compared to males and individuals with lower BMI, respectively (*p* < 0.05). A total of 25.3% of our participants were labeled as “overweight”, while 20.4% had been overweight in the past. Significant differences were reported between body-esteem and social physique anxiety levels (*p* < 0.001); age (*p* = 0.001); BMI value (*p* < 0.001) and never having a problem with body weight (*p* = 0.008). Additionally, individuals with lower body-esteem and greater social physique anxiety levels presented lower global self-esteem (*p* < 0.001). (4) Conclusions: individuals’ engagement in physical activity promotes physical as well as mental well-being, contributing to an improved quality of life, which may be the most important issue for health care professionals.

## 1. Introduction

A preoccupation with body image and physical appearance has been observed in humans since ancient times. “A sound mind in a sound body” is the English translation of a famous quote by the pre-Socratic Greek philosopher Thales (Miletus, 624–546 BC), which demonstrates that physical exercise, mental balance and the capacity for pleasure in life are closely related. Individuals need to be involved in some kind of physical and athletic activity in order to promote their physical and mental well-being. It has been found that increased levels of endorphins and neurotransmitters (noradrenaline, serotonin and dopamine) stimulated by physical activity may raise an individual’s mood and euphoria [1,2]. Additionally, catecholamine levels are increased, cortisol levels and systemic inflammation are decreased and the functioning of the hypothalamus-pituitary-adrenal (HPA) axis is improved by regular physical activity [3,4,5].

A great number of individuals may be engaged in exercise and physical activity programs in order to improve their body appearance and increase their perception of social acceptance [6,7]. A high positive correlation of participating in physical activity programs with body image and self-esteem has been observed. The improvement of body appearance could result in higher self-esteem, which is defined as the global attitude of individuals regarding their positive or negative evaluations about themselves [8].

Body image refers to an individual’s subjective perception of their physical appearance, including their size, shape and overall attractiveness. It also encompasses individuals’ attitudes, beliefs and emotions related to their body and how they perceive others’ perceptions of their body. Consequently, body image could also be directly associated with social physique anxiety levels. Social physique anxiety has been defined as an emotional state that involves individuals’ concerns about the way that their body is perceived by other people. Their beliefs about being poorly judged for their body structure, shape, height and weight may cause them great worry, embarrassment or shame [9]. Exposing their bodies in gyms and other athletic environments could make them feel frustrated, angry, nervous or embarrassed [10]. Social physique anxiety could be considered as a form of social anxiety. As a result, individuals may choose to either engage in or avoid physical activity programs in order to avoid their bodies being exposed and negatively perceived by others [11]. 

Social physique anxiety has been positively associated with body image dissatisfaction and lower self-esteem [12] and negatively associated with being engaged in physical activity programs [9]. Individuals may be distressed due to social criticism related to their physical features; they may also consider their body as being unattractive. In the Greek population, social physique anxiety has been reported to be influenced by parents’ pressure on their teenage children in order to have a well-shaped and fit body [13]. 

The aim of this study was to investigate body image, BMI characteristics, social physique anxiety, self-esteem and possible correlations between the above factors in individuals participating in exercise programs and athletic activities. This study in the Greek population adds to the previous research by investigating the specific associations in this cultural context. By understanding the cultural and societal influences on individuals’ body image and physical self-perception, the study may be able to inform the development of more effective and culturally appropriate strategies to address them.

## 2. Materials and Methods

### 2.1. Study

A cross-sectional study was conducted from February to May 2020 in gyms and athletic facilities in the Municipality of Elefsina, which is located in the West Attica regional area of Greece. The sampling method applied was that of convenience sampling. A total of 262 questionnaires were distributed and 245 were completed (response rate: 93.51%).

### 2.2. Participants

The sample of the study consisted of 245 adults engaged in training programs in gyms, as well as in track and field, football and basketball athletic activities. The inclusion criteria were (i) age > 18 years, (ii) ability to read and sign the consent form and (iii) ability to speak, read and write in Greek. Minors, individuals who did not wish to participate and those who had inadequate language skills were excluded from the study.

### 2.3. Instruments

The first part of the self-report questionnaire consisted of questions to collect information related to the sociodemographic characteristics of the respondents (gender, marital status, children, level of education and professional status), possible previous problems with body weight, type of exercise, frequency and main reasons for exercising, as well as the somatometric characteristics of the participants (age, height and body weight). BMI was computed for each participant as weight (kg) divided by squared height (m^2^). The BMI ranges were categorized as follows: (a) underweight: BMI less than 18.5; (b) normal weight: BMI between 18.5 and 25; (c) overweight: BMI between 25 and 30; (d) obese: BMI of 30 or greater. Respondents were classified into normal weight, overweight and obese individuals, according to their BMI index. It is important to note that while BMI is a widely used and useful screening tool for assessing weight-related health risks, it has its limitations and should not be the sole determinant of an individual’s health status [14]. The Body-Esteem Scale for Adolescents and Adults of Mendelson et al. [15] was used to assess body image in the second part of the questionnaire. The scale consists of 23 questions and has been translated into the Greek language and validated in the Greek population [16]. A higher score is indicative of higher body-esteem. The questionnaire is divided into three subscales. Participants respond on a 5-point Likert scale (0 = never and 4 = always). The scores of the negatively worded items must be reversed. The first subscale (“Appearance”) consists of 10 questions with 4 positively and 6 negatively worded items and assesses individuals’ general feelings about their appearance. The second subscale (“Weight”) consists of 5 positively and 3 negatively worded questions (3, 4, 8, 10, 16, 18, 19 and 21) and evaluates the weight satisfaction of the respondent. The third subscale (“Attribution”) consists of 5 positively worded questions (2, 5, 12, 14 and 23) and includes evaluations attributed to others about one’s body and appearance. The three subscales have high internal consistency (Cronbach’s alpha of 0.92, 0.94 and 0.81) and high test–retest reliability (r = 0.83–0.92) [15].

The third part was derived from the Rosenberg Self-Esteem Scale [17], which is a reliable psychometric instrument for assessing global self-esteem. The scale consists of 10 items, with 5 (1, 3, 4, 7 and 10) positively and 5 (2, 5, 6, 8 and 9) negatively worded. Participants respond on a 4-point Likert scale, ranging from 1 “strongly disagree”, to 2 “disagree”, to 3 “agree” to 4 “strongly agree”. The total score of the scale, after reversing the scores of the negatively worded items, is indicative of an individual’s level of global self-esteem, with higher scores reflecting higher levels of global self-esteem. The scale has also been used in the Greek population with high validity and reliability [18,19].

The fourth part of the questionnaire included the Social Physique Anxiety Scale (SPAS) [20] to assess the degree to which people become anxious when others observe or evaluate their physical appearance. The scale consists of 12 questions and participants respond on a 5-point Likert scale, ranging from 1 (not at all) to 5 (very much). Higher scores on the scale, after reversing the scores of the negatively worded items (1, 5, 8 and 11), are indicative of a higher level of stress. The Social Physique Anxiety Scale demonstrates both high internal and test–retest reliability. The Greek version of the scale [21] has high internal consistency (Cronbach’s a 0.85 and high test–retest reliability [r = 0.84; (*p* < 0.01)]).

### 2.4. Statistical Analysis

All statistical analyses were performed using the IBM Statistical Package for Social Sciences (SPSS Inc., Chicago, IL, USA), version 21.0. Sociodemographic data were analyzed using descriptive statistics. Frequencies and percentages were calculated for the categorical variables, while continuous variables were expressed as the mean (standard deviation (SD)) or the median (interquartile range). The Kolmogorov–Smirnov test and normality plots were used to test for the normal distribution of continuous data. The *t*-test, Pearson correlation coefficient, Spearman’s correlation coefficient and multiple linear regression analysis were used. Statistical significance was set at 0.05. It is noted that for the selection of the independent variables entered into the multivariate logistic regression models, the value *p* < 0.2 was used as a limit in order to limit the possibility of not including independent variables in the models that were related to the dependent variables, as much as possible. With reference to the multiple linear regression, the standardized coefficients’ beta, the corresponding 95% confidence intervals and the *p*-values are presented. Internal consistency for the questionnaire was evaluated using Cronbach’s alpha indexes. Values ≥ 0.7 were indicative of good internal consistency of the items. 

### 2.5. Ethical Considerations

Before collecting data, approval was obtained by the scientific boards of each athletic facility involved (approval numbers: 01-8/01/2020; 01-10/01/2020; 02-30/01/2020 and 04-17/02/2020). All participants were informed about the purpose of the study, the confidentiality of their responses, the anonymity of the data and the voluntary nature of their participation, as well as their right to refuse or discontinue participation in the study. They were also informed that the data would be used only for research purposes. All participants participated only after they had given their written consent. The research complied with the General Regulation for the Protection of Personal Data (GDPR) and the Declaration of Helsinki (1989) of the World Medical Association. 

## 3. Results

The sample consisted of 245 participants (62.9% were male and 37.1% were female), with a mean age of 35.0 years (SD = 12.0). The majority of the participants were single (54.3%), had no children (60.0%) and had had a university education (50.6%) (Table 1). 

The somatometric characteristics of the total sample by gender are shown in Table 2, while the classification of the sample according to BMI is shown in Table 3.

The frequency and percentage distribution of the sample according to smoking, drinking and exercise habits (type and frequency of exercise and reason for exercising) are shown in Table 4.

Differences between participants are shown in Table 5. More specific results are detailed below:Individuals who exercised in a gym exercised less often (in times/week) than those who played football, basketball or were involved in track and field activities (*p* = 0.001).People who exercised in a gym had a lower score on the “Appearance” subscale (*p* < 0.001), “Weight” subscale (*p* < 0.001) and “Attribution” subscale (*p* = 0.013) and had a lower Body-Esteem score (*p* < 0.001) and Self-Esteem score (*p* = 0.007) than people who played football, basketball or were involved in track and field activities.Individuals who exercised in a gym had a higher Social Physique Anxiety Scale score than individuals who played football, basketball or were involved in track and field activities (*p* < 0.001).Individuals who exercised in a gym were more likely to exercise to lose weight and improve their appearance than people who played football, basketball or were involved in track and field activities (*p* < 0.001).

In the current study, Cronbach’s alpha for the Body-Esteem Scale for Adolescents and Adults was reported to be 0.943, while Cronbach’s alpha for the “Appearance”, “Weight” and “Attribution” subscale was 0.904, 0.924 and 0.776, respectively. Additionally, Cronbach’s alpha for the Rosenberg Self-Esteem Scale and Social Physique Anxiety Scale was reported to be 0.806 and 0,846, respectively. All Cronbach’s alphas indicated a high reliability of the scales in this study.

Women had a lower score on the “Appearance” subscale (*p* = 0.046) and a higher Social Physique Anxiety Scale score than men (*p* = 0.002) (Table 6).

Men exercised competitively to a greater extent than women (for athletic competitions), while women exercised to a greater extent than men for appearance improvement and health reasons (Table 7).

According to the multivariate linear regression, the following results are presented (Table 8):Age increase was related to an increase in the “Appearance” subscale score (*p* = 0.030), an increase in the “Weight” subscale score (*p* = 0.018) and a decrease in the Body-Esteem score (*p* = 0.001).An increase in BMI was associated with a decrease in the “Appearance” subscale score (*p* < 0.001), a decrease in the “Weight” subscale score (*p* < 0.001), a decrease in the “Attribution” subscale score (*p* < 0.001), a decrease in the Body-Esteem score (*p* < 0.001) and an increase in the Social Physique Anxiety Scale score (*p* < 0.001).An increase in the Social Physique Anxiety Scale score was associated with a decrease in the Body-Esteem score (*p* < 0.001).Women had a higher Social Physique Anxiety Scale score than men (*p* = 0.003).An increase in the time period (in years) that individuals were at their current weight for was related to an increase in the score of the “Weight” subscale (*p* = 0.002).Individuals that had never had a problem with their body weight had a higher “Weight” subscale score (*p* = 0.041) and a higher Body-Esteem score (*p* = 0.008) compared to individuals with a higher body weight.Employed individuals had a lower “Attribution” subscale score than unemployed individuals (*p* = 0.041).An increase in the Self-Esteem score was related to an increase in the Body-Esteem score (*p* < 0.001), an increase in the “Appearance” subscale score (*p* < 0.001), an increase in the “Weight” subscale score (*p* < 0.001) and an increase in the “Attribution” subscale score (*p* < 0.001).An increase in the Social Physique Anxiety Scale score was related to a decrease in the Self-Esteem score (*p* < 0.001).

## 4. Discussion

Physical exercise has a positive effect on individuals’ physical and mental health, which could be extremely important if we take into consideration the increased pressure applied to keep a balance of domestic and work responsibilities. It is remarkable that 46.1% of participants in the present study reported being engaged in exercise programs for pleasure. Similar findings were reported in the study conducted by Asztalos et al. [22]. Physical activity may raise individuals’ moods due to the increase in endorphins which, along with endogenous opioid peptides, provide a state of euphoria and pleasure [1]. Catecholamine levels are increased, cortisol levels and systemic inflammation are decreased and the functioning of the hypothalamus-pituitary-adrenal (HPA) axis is improved by regular physical activity [3,4,5]. Aminergenic synaptic transmission (noradrenaline, serotonin and dopamine) has been reported to be stimulated by exercise. Increased levels of the aforementioned neurotransmitters could have a positive effect on mood [2]. No complete explanations have been given with regard to these theories; further research is required.

The mean age of those attending the gym was 35 years of age (SD = 12.0). A total of 70% worked out at the gym 3–5 times a week and 73.1% spent 2 h a day at the gym. Young people who are engaged in exercise programs may be committed to a healthier way of life [23]. Only 15.9% and 14.3% of the participants were actively smoking and drinking alcohol, respectively. Low percentages of the participants seemed to adopt these unhealthy habits. Being engaged in exercise and sports could be a protective factor against tobacco and alcohol use, leading to the adoption of healthier behaviors [24].

With the exception of the 46.1% of participants who reported exercising for pleasure, 20.8% exercised for athletic competitions, 18.4% exercised for health reasons, 7.8% exercised to lose weight and 6.9% exercised to improve their appearance. Both genders tended to work out to improve their physical health by participating in competitive sports/athletic competitions [25,26], to improve their body appearance or to lose weight [27,28].

Individuals who worked out at the gym had lower scores in the “Weight”, “Appearance” and “Attribution” subscales compared to those who were involved in football, basketball or track and field activities. This means that people who worked out at the gym were less satisfied with their body weight and appearance and also had lower evaluations attributed to others about their bodies compared to people who participated in the aforementioned athletic activities. It is important to note that a large percentage of our participants who engaged in athletic activities had professional careers in those sports, while those engaged in gym training mainly wanted to become fit and maintain their good shape. Athletes are usually distinguished for their competitive spirit as well as their strong will, discipline, self-control and obedience regarding their exercise and nutrition programs [29]. More than 60% of elite athletes from lean-focused and non-lean-focused sports reported pressure from coaches concerning body shape [30]. This could possibly explain the greater body satisfaction of the participants engaged in athletic activities compared to those who worked out at the gym, whose lower satisfaction could, in turn, explain their lower evaluations attributed to others about their body appearance as well as their higher social physique anxiety. In addition, indoor aerobic activities that take place in a gym usually take place in rooms with mirrors, where participants’ body exposure to the eyes of other men and women exercising at the same time contribute to additional social physique anxiety, especially for those who are overweight. Another interpretation of the above results may be due to the fact that the majority (71%) of those exercising in the gym were women, who reported statistically significant lower evaluations regarding their body appearance compared to men. Women seemed to pay more attention to body weight and attractiveness when compared to men [31]. Additionally, Nanu et al. [32], who studied a sample of young people aged 15–20 years, concluded that boys were more satisfied with their body appearance and weight compared to girls. Gender differences can be explained by the fact that girls’ and women’s identities can be more associated with body image, due to their compliance with social representations which promote the correlation of female physical attractiveness with self-worth. Women are more likely to be criticized harshly for their body image [33,34,35]. Female body attractiveness is emphasized by advertisements, mass media, magazines and social media [36].

As a result, women also reported higher social physique anxiety compared to men, meaning that they were more anxious than men because of others’ evaluations and judgements regarding their bodies. This finding was also reported in the study by Mack et al. [37]. Additionally, female anxiety could be increased by biological factors. Female gonadal hormones, such as estrogen and progesterone, seem to impact substantially on the function of anxiety-related neurotransmitter systems and affect fear elimination [38,39]. On the other hand, testosterone has been reported to have an anxiolytic effect, possibly by reducing the responsiveness to stress and suppressing the activity of the hypothalamic pituitary adrenal (HPA) axis [40]. Consequently, at least part of the increased prevalence of anxiety disorders in females could be explained due to gonadal hormones’ function.

Girls have learned to be praised or judged for their appearance from the beginning of their socialization process. In a study of 138 women (European and American) aged from 40 to 87 years, it was found that age did not alleviate women’s special interest in or dissatisfaction with their body [41]. According to social norms, the female body may be considered of great importance for a woman in order to be attractive to the opposite sex or in finding a sexual partner [42]. It is for all of the above reasons that women may have reported the improvement in their appearance as a reason for exercising more than men. Individuals suffering from higher social physique anxiety, such as women, seemed to choose exercising to improve their appearance [22].

Since people who exercised in the gym were less self-satisfied with their body weight, it was expected that they exercised in order to lose weight to a greater extent than those who were involved in football, basketball or track and field activities in the present study. One of the main reasons for exercising in the gym is controlling body weight and improving appearance [43]. The above finding is likely to be explained, also, by the fact that the majority (71%) of those exercising in the gym were women, who reported that they exercised to improve their appearance at a higher percentage than men according to the results of the study.

On the contrary, men were more satisfied with their appearance compared to women. Similar results have been reported in the study of Zaccagni and Gualdi-Russo [44]. For boys and men, the social norms are not so strict. Although the media is more likely to present the ideal of a muscular and athletic male type, there are different, but still socially accepted, types of men (with different physical characteristics) that women find attractive. Additionally, although men may be concerned about their body performance, they tend to internalize their concerns as they may not be socially allowed to talk “openly” about them [45].

Compared to women, more men were reported to engage in exercise competitively. It is also worth noting that girls in adolescence tend to have celebrities as role models, in contrast with boys whose role models are mostly athletes. Men, compared to women, are generally more competitive in many areas of their social life and consequently in sports [29]. As a result, it is possible that men develop more skills, abilities and self-confidence; consequently, they may experience statistically significant lower social physique anxiety compared to women [46].

Moreover, individuals who were involved in athletic activities reported lower social physique anxiety than those who exercised in the gym, perhaps due to feelings of intimacy between team members which, in turn, helped to improve team cohesion [47]. 

According to our results, the increase in global Self-Esteem score was associated with an increase in Body-Esteem score. Additionally, people who reported lower satisfaction with their body weight reported lower self-esteem. Body image dissatisfaction may lead to lower self-esteem [48,49]. These findings support a correlation between self- and body-esteem which may be due to the association of attractiveness, success and popularity with a fit body according to social standards. Similar findings have also been reported in a study by Duchesne et al. [50]. On the other hand, low self-esteem could also be a predictor of body dissatisfaction [51,52]. When low self-esteem persists for a long time, individuals may become more vulnerable when compared with social standards of the ideal body, resulting in increased body dissatisfaction [53].

On the contrary, people who rated their appearance as being higher showed higher self-esteem as well. In addition, when individuals believe that others perceive their appearance to be more attractive, they tend to have lower social physique anxiety and higher self-esteem. These findings are supported by other studies [12,54,55]. Finally, considering that people who exercised in the gym showed lower body-esteem, it is obvious why they reported lower self-esteem as well compared to people who participated in organized sports, considering the relationship between body- and self-esteem.

People with a higher BMI showed statistically significant lower satisfaction with their body weight and appearance, as well as lower body-esteem. BMI is a biological indicator for calculating a person’s degree of obesity. A total of 25.3% of the participants had a BMI of 25–30, meaning that they were considered to be overweight. The vast majority of participants weighed within normal limits. BMI could determine the degree of satisfaction with body image, size and shape [56,57]. In particular, a high BMI has been associated with higher physical dissatisfaction [58,59] and increased concerns related to weight among young people [60,61]. A negative correlation has been recorded between body weight and “body-esteem” [62]. Obesity have been positively associated with an increased risk of body dissatisfaction and low self-esteem among children and adolescents [63].

People with a higher BMI had lower body-esteem, resulting in higher social physique anxiety levels, which is compatible with Mack et al.’s [47] study. Obesity and anxiety disorders have been positively associated with one another in both men and women [64]. Obesity is regarded as a mild form of chronic inflammation. Adipose tissue is acknowledged to produce a great amount of pro-inflammatory cytokines and is considered to be a link between inflammation and obesity [65]. Anxiety has been correlated with increased inflammation in a great number of studies [66,67,68]. Those who are obese and socially anxious may be more susceptible to increased inflammation and insulin resistance compared to those who are characterized by obesity but not social anxiety, due to the interaction of the aforementioned factors which can be extremely stressful [69]. 

A total of 20.4% of the sample had experienced a problem with being overweight at some point in the past. People who had maintained their current weight for a longer period of time showed higher satisfaction with their body weight. In addition, people who had never had a weight problem showed higher body-esteem compared to people who had had a weight problem in the past. One’s experience of losing and regaining weight may lead to greater frustration about their body than those experiencing stable weight [70]. The psychological effects of this vicious cycle could lead to a negative body image. In a study by Friedman et al. [71], there was a distinction between an individual’s subjective perception regarding losing and regaining weight and the objective number of cases of weight fluctuation. There are indications that after weight loss, body image problems persist, a fact that was called “phantom fat” by Milkewicz and Cash [72], who investigated whether overweight, ex-overweight and never-overweight individuals differed in relation to their body image. People who were previously overweight did not eventually have as much of a positive body image when they lost weight as someone who had never been overweight. Instead of seeing a fit body in the mirror, they perceived themselves as still being very heavy. The perception of body image is more of a cognitive issue than a physical one. According to the “Allocentric Lock Hypothesis” [73], individuals may be locked into a negative “objectified body” which is created by a public representation of their own body by what others see and judge, and cannot be updated or modified by contrasting representations driven by their own perceptions, even after a significant weight loss. Exogenous and/or endogenous stress mechanisms could impact substantially in the functioning of the medial temporal lobe and induce a locked negative body image. Anxiety and stress could make worse the structurally and/or functionally vulnerable brain regions involved in the egocentric/allocentric encoding process, which could possibly explain the above findings [73]. The hippocampus substantially influences spatial memory by developing allocentric representations for long-term memory [74]. Stress may harm CA3 hippocampal neurons [75,76], which could in turn disrupt hypothalamic-pituitary-adrenal (HPA) axis activity, resulting in dysregulated glucocorticoid release, which, combined with hippocampal CA3 dendritic retraction, could impair spatial memory, including the bodily one [75]. 

Older participants showed lower satisfaction with their body weight and lower body-esteem compared to younger participants. As people get older, they may begin to reduce their exercise time and increase their sedentary lifestyle, causing an increase in body weight [77,78]. Additionally, as people get older, their basic metabolism decreases. In addition, muscle shrinks and the percentage of fat increases. Exercising decreases and muscle atrophy develops as a result of reduced mobility, which makes it difficult to burn calories and promotes the accumulation of fat in the body [78]. The accumulation of excess fat in the body is either due to overeating or hormonal problems and metabolic disorders. When a combination of increased calorie intake, decreased body exercise and decreased metabolic needs occurs, then the observed hyper-lipogenesis contributes to weight gain [79]. Aging is also associated with increased and persistent low-level inflammation [80] and possibly with obesity, which is associated with an increased proinflammatory state [81]. The aforementioned findings need to be treated with appropriate caution due to the limitations of the present study.

### Limitations

Our sample included trainees from athletic clubs and gyms located in a single region of Attica. Therefore, the generalizability of the results is limited. The use of self-report questionnaires is also a limitation. Even though they included some validated measures, there was no corresponding clinical interview. There is a possibility that answers were given based on what is socially correct and desirable. The study had a cross-sectional design; so, causal associations cannot be drawn from the findings. Finally, although the unit of measurement of height was centimeters (cm), data were obtained in meters (m) for BMI calculation purposes. In the light of the exploratory nature of the present study, the authors have a priori decided to present all findings arising from various individual statistical tests while maintaining a statistically significant level of *p* < 0.05. However, it could be argued that there should be a provision for inflated type I error (false positive findings). On the other hand, as mentioned earlier, our aim was to unearth any correlations between the concepts under examination, without having set any endpoint(s). Therefore, by applying correction techniques to avoid a type I error, critical findings would have been left out from our study. It is our intention to highlight every possible contributing factor for the interested researchers in order to take them into account and formulate a specific hypothesis(/es) to verify each of them. Finally, future research could be extended to individuals not engaged in physical activity programs to investigate body image, self-esteem and social physique anxiety so that comparisons can be made between exercisers and non-exercisers.

## 5. Conclusions

Physical activity may increase the production and release of neurotransmitters and stimulate certain areas of the brain, such as the hippocampus, which is involved in learning and memory. Physical activity could promote the growth of new neurons in this area, leading to improvements in memory and cognitive function, while dopamine, serotonin and endorphins could be involved in mood regulation and stress response. These effects could lead to improvements in mood, cognitive function and overall well-being. On the other hand, the social representations of women and men related to body image, as well as differences in gonadal hormones, could contribute to the relative prevalence of anxiety, especially amongst women. Addressing these issues through education and awareness campaigns, as well as through personalized treatment approaches, could help to mitigate the negative impact on mental health. The findings of the present study could be used by professionals working in the athletic and training fields, as well as health professionals, in order to demonstrate greater awareness, empathy and effectiveness in handling body image and self-esteem issues in both genders. The implementation of strategies that reduce body dissatisfaction and increase self-esteem could reduce the risk of developing eating and emotional disorders. The findings of this study could enrich the knowledge for a deeper understanding of the relationship between engagement in exercise programs and body image, in order to improve outcomes and promote a higher quality of life. Future research could explore the long-term effects of physical activity on the brain, as well as the effects of various types of exercise on mood and neurobiology in more detail. Future research could also use advanced neuroimaging techniques to better understand how physical activity affects different brain regions and the pathways involved in mood regulation.

## Figures and Tables

**Table 1 diseases-11-00066-t001:** Sociodemographic characteristics of the sample (Ν = 245).

		N	%
**Gender**	Male	154	62.9
	Female	91	37.1
**Marital status**	Single	133	54.3
	Married	103	42.0
	Divorced	8	3.3
	Widowed	1	0.4
**Children**	Yes	98	40.0
	No	147	60.0
**Education**	Secondary education	1	0.4
	High school	84	34.3
	Technological education/University education	124	50.6
	MSc	26	10.6
	PhD	2	0.8
	Other	8	3.3
**Occupation**	Freelancer	43	17.6
	Civil servant	55	22.4
	Private sector employee	74	30.2
	Student	39	15.9
	Unemployed	16	6.5
	Pensioner	6	2.5
	Household	2	0.8
	Other	10	4.1

**Table 2 diseases-11-00066-t002:** Somatometric characteristics of the sample (Ν = 245) by gender.

Somatometric Characteristics	Male	Female	Total
	Mean (SD)	Mean (SD)	Mean (SD)
**Height (in meters)**	1.82 (0.07)	1.68 (0.06)	1.77 (0.10)
**Weight (in kg)**	80.77 (10.27)	63.81 (9.21)	74.5 (12.8)
**BMI**	24.24 (2.2)	22.41 (2.47)	23.6 (2.5)
**How long have you been your current weight for (in years)? ^a^**	5.31 (5.3) ^a^	5.37 (5.28) ^a^	3 (6.0) ^a^
	**Ν (%)**	**Ν (%)**	**Ν (%)**
**Have you ever had a problem with your body weight?**			
Yes	38 (24.7)	19 (20.9)	57 (23.3)
No	116 (75.3)	72 (79.1)	188 (76.7)
**If yes:**			
I was overweight or obese	33 (21.4)	17 (18.7)	50 (20.4)
I was underweight	5 (3.3)	2 (2.2)	7 (2.9)

^a^ Median (interquartile range)**.** SD: standard deviation. BMI: body mass index.

**Table 3 diseases-11-00066-t003:** Classification of the sample according to BMI (Ν = 245).

BMI	Ν (%)
BMI < 18.5	17 (6.9)
18.5 < BMI < 25	166 (67.8)
25 < BMI < 30	62 (25.3)
BMI > 30	0 (0.0)

BMI: body mass index.

**Table 4 diseases-11-00066-t004:** Frequency and percentage distribution of the study sample according to smoking, drinking and exercise habits (Ν = 245).

Characteristics	N (%)
**Do you smoke?**	
Yes	39 (15.9)
No	206 (84.1)
**Do you drink alcohol?**	
Yes	35 (14.3)
No	210 (85.7)
**Type of exercise**	
Gym	69 (28.2)
Football	50 (20.4)
Basketball	59 (24.1)
Track	67 (27.3)
**How many times a week do you exercise (times/week)?**	
2	16 (6.5)
3	57 (23.3)
4	66 (26.9)
5	51 (20.8)
6	46 (18.8)
7	9 (3.7)
**How many hours a day do you exercise for?**	
1	61 (24.9)
2	179 (73.1)
3	5 (2.0)
**Why do you exercise?**	
For health reasons	45 (18.4)
For championship	51 (20.8)
For pleasure	113 (46.1)
To lose weight	19 (7.8)
To improve appearance	17 (6.9)

**Table 5 diseases-11-00066-t005:** Differences between type of exercise and participants’ gender, somatometric characteristics, habits and scale and subscale scores.

Characteristics	Type of Exercise				*p*-Value
	Gym	Football	Basketball	Track	
**Gender**					**<0.001** ^a^
Male	20 (29.0%)	50 (100.0%)	41 (69.5%)	43 (64.2%)	
Female	49 (71.0%)	0 (0.0%)	18 (30.5%)	24 (35.8%)	
**BMI** ^b^	23.7 (3.1)	23.3 (1.9)	23.9 (2.3)	23.3 (2.3)	0.363 ^c^
**How long have you been your current weight for (in years)?** ^b^	3.00 (8)	2.00 (4)	3.00 (4)	5.00 (8)	0.058 ^d^
**Have you ever had a problem with your body weight?**					0.104 ^a^
Yes	19 (27.5%)	6 (12.0%)	12 (20.3%)	20 (29.9%)	
No	50 (72.5%)	44 (88.0%)	47 (79.7%)	47 (70.1%)	
**If yes:**					0.553 ^a^
I was overweight or obese	18 (94.7%)	3 (50.0%)	10 (83.3%)	19 (95.0%)	
I was underweight	1 (5.3%)	3 (50.0%)	2 (16.7%)	1 (5.0%)	
**Do you smoke?**					0.053 ^a^
Yes	18 (26.1%)	5 (10.0%)	7 (11.9%)	9 (13.4%)	
No	51 (73.9%)	45 (90.0%)	52 (88.1%)	58 (86.6%)	
**Do you drink alcohol?**					0.834 ^a^
Yes	16 (23.2%)	10 (20.0%)	14 (23.7%)	12 (17.9%)	
No	53 (76.8%)	40 (80.0%)	45 (76.3%)	55 (82.1%)	
**How many times a week do you exercise (times/week)?**					**0.001** ^a^
2	6 (8.7%)	4 (8.0%)	3 (5.1%)	3 (4.5%)	
3	34 (49.3%)	4 (8.0%)	9 (15.3%)	10 (14.9%)	
4	16 (23.2%)	12 (24.0%)	16 (27.1%)	22 (32.8%)	
5	10 (14.5%)	9 (18.0%)	17 (28.8%)	15 (22.4%)	
6	3 (4.3%)	16 (32.0%)	14 (23.7%)	13 (19.4%)	
7	0 (0.0%)	5 (10.0%)	0 (0.0%)	4 (6.0%)	
**How many hours a day do you exercise for?**					0.313 ^a^
1	32 (46.4%)	3 (6.0%)	4 (6.8%)	22 (32.8%)	
2	37 (53.6%)	46 (92.0%)	52 (88.1%)	44 (65.7%)	
3	0 (0.0%)	1 (2.0%)	3 (5.1%)	1 (1.5%)	
**Why do you exercise?**					**<0.001** ^a^
For health reasons	16 (23.2%)	4 (8.0%)	7 (11.9%)	18 (26.9%)	
For championship	1 (1.4%)	25 (50.0%)	17 (28.8%)	8 (11.9%)	
For pleasure	29 (42.0%)	19 (38.0%)	30 (50.8%)	35 (52.2%)	
To lose weight	11 (15.9%)	1 (2.0%)	2 (3.4%)	5 (7.5%)	
To improve appearance	12 (17.5%)	1 (2.0%)	3 (5.1%)	1 (1.5%)	
**“Appearance” subscale score** ^b^	26.6 (6.5)	32.1 (5.5)	30.2 (6.1)	31.3 (5.2)	**<0.001**
**“Weight” subscale score** ^b^	19.1 (6.5)	24.7 (5.8)	22.6 (5.6)	24.1 (5.7)	**<0.001**
**“Attribution” subscale score** ^b^	12.1 (3.2)	14.0 (3.2)	13.3 (3.0)	13.2 (3.0)	**0.013**
**Body-Esteem score** ^b^	57.8 (14.0)	70.8 (13.0)	66.0 (12.8)	68.6 (11.8)	**<0.001**
**Self-Esteem Scale score** ^b^	33.1 (3.3)	34.4 (3.1)	34.5 (2.9)	34.9 (3.2)	**0.007**
**Social Physique Anxiety Scale score** ^b^	28.3 (7.6)	22.4 (5.3)	24.2 (5.7)	22.3 (5.6)	**<0.001**

^a^ Chi-square test. ^b^ Mean value (SD: standard deviation). ^c^ Analysis of variance. ^d^ Kruskal–Wallis test. BMI: body mass index.

**Table 6 diseases-11-00066-t006:** Results of *t*-test for comparison of scales and subscales by gender.

	Gender						
	Male		Female				
	Mean	SD	Mean	SD	t	df	*p* *
“Appearance” subscale score	30.5	6.1	28.8	6.3	2.003	243	**0.046**
“Weight” subscale score	22.8	6.1	21.8	6.6	1.278	243	0.203
“Attribution” subscale score	13.0	3.1	13.2	3.2	−0.316	243	0.752
Body-Esteem score	66.4	13.5	63.8	14.3	1.412	243	0.159
Self-Esteem Scale score	34.3	3.0	34.2	3.5	0.297	243	0.767
Social Physique Anxiety Scale score	23.5	6.0	26.2	7.4	−3.105	243	**0.002**

*: *t* test. SD: standard deviation.

**Table 7 diseases-11-00066-t007:** Results of chi-square test for comparison of previous weight problems and reasons for exercising by gender.

Characteristic	Gender		*p*-Value
	Male	Female	
**Have you ever had a problem with your body weight?**			0.497 ^a^
Yes	38 (24.7%)	19 (20.9%)	
No	116 (75.3%)	72 (79.1%)	
**If yes:**			0.571 ^a^
I was overweight or obese	33 (86.8%)	17 (89.5%)	
I was underweight	5 (13.2%)	2 (10.5%)	
**Why do you exercise?**			**0.050** ^a^
For health reasons	25 (16.2%)	20 (22.0%)	
For championship	39 (25.3%)	12 (13.1%)	
For pleasure	73 (47.4%)	40 (44.0%)	
To lose weight	10 (6.5%)	9 (9.9%)	
To improve appearance	7 (4.6%)	10 (11.0%)	

^a^ Chi-square test.

**Table 8 diseases-11-00066-t008:** Results of multivariate linear regression analysis (Ν = 245).

**Multivariate linear regression with the “Appearance” subscale score as the dependent variable**				
**Independent variables**	Coefficient b (standardized)	Adjusted R^2^	95% confidence interval for b	*p* value
Age (in years)	−0.148	0.310	−0.073 to −0.004	**0.030**
BMI	−0.278		−0.419 to −0.227	**<0.001**
**Multivariate linear regression with the “Weight” subscale score as the dependent variable**				
**Independent variables**	Coefficient b (standardized)	Adjusted R^2^	95% confidence interval for b	*p* value
Age (in years)	−0.187	0.387	−0.092 to −0.009	**0.018**
BMI	−0.405		−0.414 to −0.214	**<0.001**
How long have you been your current weight for (in years)?	0.204		0.050 to 0.230	**0.002**
Have you ever had a problem with your body weight?	0.238		0.047 to 2.239	**0.041**
**Multivariate linear regression with the “Attribution” subscale score as the dependent variable**				
**Independent variables**	Coefficient b (standardized)	Adjusted R^2^	95% confidence interval for b	*p* value
Occupation	0.121	0.147	0.034 to 1.541	**0.041**
BMI	−0.174		−0.161 to −0.39	**<0.001**
**Multivariate linear regression with the Body-Esteem score as the dependent variable**				
**Independent variables**	Coefficient b (standardized)	Adjusted R^2^	95% confidence interval for b	*p* value
Self-Esteem Scale score	0.216	0.722	0.588 to 1.282	**<0.001**
Social Physique Anxiety Scale score	−0.569		−1.356 to −1.005	**<0.001**
Age (in years)	−0.118		−0.215 to −0.057	**0.001**
BMI	−0.220		−1.641 to −0.814	**<0.001**
Have you ever had a problem with your body weight?	−0.101		−0.861 to −5.742	**0.008**
**Multivariate linear regression with the Self-Esteem Scale score as the dependent variable**				
**Independent variables**	Coefficient b (standardized)	Adjusted R^2^	95% confidence interval for b	*p* value
“Appearance” subscale score	0.452	0.440	0.140 to 0.319	**<0.001**
“Weight” subscale score	−0.312		0.056 to 0.217	**<0.001**
“Attribution” subscale score	0.265		0.152 to 0.385	**<0.001**
Social Physique Anxiety Scale score	−0.297		−0.216 to −0.067	**<0.001**
**Multivariate linear regression with the Social Physique Anxiety Scale score as the dependent variable**				
**Independent variables**	Coefficient b (standardized)	Adjusted R^2^	95% confidence interval for b	*p* value
Gender	0.324	0.342	0.545 to 2.589	**0.003**
BMI	0.357		0.186 to 0.575	**<0.001**

BMI: body mass index. R^2^: proportion of explained variance.

## Data Availability

The datasets used and/or analyzed during the current study are available upon reasonable request from the corresponding author. The data are not publicly available due to privacy reasons.

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
