# Peer review of "Body Image, Social Physique Anxiety Levels and Self-Esteem among Adults Participating in Physical Activity Programs"

_diseases, 2023, doi:10.3390/diseases11020066_

Round 1

Reviewer 1 Report

Review by CVDolan of

Body Image, Social Physique Anxiety levels and Self-Esteem among Adults Participating in Physical Activity Programs by Zartaloudi, et al.

Comments:

The aim as stated is

The aim of this study was to investigate body image, body mass index (BMI) characteristics, social physique anxiety, self-esteem and possible correlations between the above factors in individuals participating in exercise programs and athletic activities.

The preceding text outlines that there are associations between these variables, so what does this add? I think it would be useful to elaborate on the motivation of this study. Also this study concerns individual who were recruited at gyms. So that implies a selected sample. Why is this group, highly educated persons (62%) who attend gyms and sportschools (100%), who engage in sports at least 4 times a week (68%),  of interest specifically?  The selection also implies restriction of range. Is that a problem?

A cross-sectional study was conducted from February to May 2020 in gyms and athletic facilities of Municipality of Elefsina, which is located in the West Attica regional unit 70 of Greece and has been announced to be the European capital of Culture for 2023.” Elefsina is capital of something. That is great, but why is it relevant info here?

The first subscale ("Appearance") 95 consists of 10 questions (1, 6, 7, 9, 11, 13, 15, 17, 20, 22)

The item numbers are mentioned. Why is that relevant information. I understand thet 10 of the 23 are devoted to Appearance. Why do I need known that these are items 1,6 etc.?

The statistical methods section includes the statement: “Statistical significance was set up at 0.05.”. This paper contains many statistical tests, so is .05 the family-wise alpha (e.g. Bonferroni corrected), or the per test alpha? If the latter, what about the  multiple testing burden?

In case that >2 independent variables presented a p-value < .20 at the bivariate analysis, multivariate linear regression was applied with subscale and total scale scores as the dependent variable. In this case, backward stepwise linear regression was performed ...”

I cannot follow this. Please rephrase. Why p<.20 exactly? multiple regression is done with backward stepwise regression. So that means that the aim is to determine the best subset of predictors in a set of predictors. What are the predictors? what is the dependent? Why focus on the best subset? Why not report results for all predictors as obtained in the regression analyses? What statistical criteria were used (F to remove etc.)? 

The results concerning Cronbach’s alpha are reported in the stat method section. Don’t these belong in the results section? I never understand why Cronbach alpha's are reported of well developed psychometric instruments. 

According to the multivariate linear regression, the following results are presented (Table 8):” It would be useful to state explicitly the full set of predictors. I would prefer to see the regression coefficients of the full set (rather than a subset).

The regression results (Table 8) include the raw regression coefficients. It would be useful to include the standardized regression coefficients. Also for each analyses, please include the R^2 (proportion of explained variance). To understand the effect size, we require standardized betas and R^2. The effect sizes (R^2) are also important for the discussion of the implications of the results. If the R^2 are low, then that has a bearing on the implications.

An alternative approach to the analysis would be to fit a structural equation model, which can accommodate the dual role of Weight, Appearance, and Attribution as dependent and independent variables. 

Minor:

The text would benefit from editing. Just a few examples:

main reasons for being exercised” ... main reason for exercising.

Respondents 87 have been classified into normal weight, overweight and obese individuals, according to 88 their BMI index.” ... were classified.

Men exercised to a greater extent than women for championships

Men exercised competitively to a greater extent than women

Higher percentage of men compared to women reported being exercised for cham- 326 pionship reasons.

Compared to women, more men reported to engage in excerise competitively.  

Author Response

Response to Reviewer 1 Comments

Dear reviewer  

We would like to thank you for taking the necessary time and effort to review the manuscript. We sincerely appreciate all your valuable comments and suggestions, which helped us in improving the quality of the manuscript.

Point 1: The preceding text outlines that there are associations between these variables, so what does this add? I think it would be useful to elaborate on the motivation of this study.

Response 1: The motivation behind this study was to further investigate these associations and their potential causes in the Greek population. This study in the Greek population builds upon the previous research by investigating the specific associations in this cultural context. By understanding the cultural and societal influences on individuals' body image and physical self-perception, the study may be able to inform the development of more effective and culturally appropriate strategies to address them. (added in line 73-77)

Point 2: Also, this study concerns individual who were recruited at gyms. So that implies a selected sample. Why is this group, highly educated persons (62%) who attend gyms and sportschools (100%), who engage in sports at least 4 times a week (68%), of interest specifically?  The selection also implies restriction of range. Is that a problem?

Response 2: I hope that I have understood this question appropriately and provided a response that meets your expectations.

The group of individuals recruited at gyms in the above study may be of interest for several reasons. Firstly, individuals who attend gyms and sports schools are likely to be more physically active than the general population, which makes them an important target group for interventions to promote positive body image and engagement in physical activity. Additionally, individuals who attend gyms and sports schools may be more likely to experience social physique anxiety due to the focus on physical appearance in these settings, which makes them an important group to study in order to understand the factors that contribute to this issue. However, the fact that the study recruited individuals exclusively from gyms and sports schools does limit the generalizability of the findings to the broader population. This restriction of range may be a problem in terms of drawing broader conclusions about the factors that contribute to social physique anxiety and body image dissatisfaction in the general population. Therefore, it is important to interpret the findings of this study with caution and to consider the limitations of the sample when applying the results to other populations.

These limitations have been referred in the Limitations section (Our sample included trainees from athletic clubs and gyms located in a single region of Attica. Therefore, the generalizability of the results is limited). Also, we have referred in line 83 that “The sampling method applied was that of convenience sampling.”

Additionally, we have added in limitations section that “Future research could be extended to individuals not being engaged in physical activity programs to investigate body image, self-esteem and social physique anxiety so that comparisons can be made between exercisers and non-exercisers.” A future study should be interested in exploring whether there are differences between individuals who attend gyms and sports schools and those who do not.

Point 3: “A cross-sectional study was conducted from February to May 2020 in gyms and athletic facilities of Municipality of Elefsina, which is located in the West Attica regional unit 70 of Greece and has been announced to be the European capital of Culture for 2023.” Elefsina is capital of something. That is great, but why is it relevant info here?

Response 3: corrected

Point 4: “The first subscale ("Appearance") 95 consists of 10 questions (1, 6, 7, 9, 11, 13, 15, 17, 20, 22)” The item numbers are mentioned. Why is that relevant information. I understand that 10 of the 23 are devoted to Appearance. Why do I need known that these are items 1,6 etc.?

Response 4: deleted

Point 5: The statistical methods section includes the statement: “Statistical significance was set up at 0.05.” This paper contains many statistical tests, so is .05 the family-wise alpha (e.g. Bonferroni corrected), or the per test alpha? If the latter, what about the multiple testing burden?

Response 5: There was no respective technique used so as to maintain a desired error rate while preserving the overall alpha level. Nevertheless, while the choice of 0.05 as the significance threshold is somewhat arbitrary, the rationale for using an alpha level of 0.05 was to balance Type I and Type II errors as a reasonable choice and due to practicality and ease of interpretation; it is simple to understand and apply and provides a reasonable trade-off between the need for evidence and the resources required to obtain that evidence.

Point 6: “In case that >2 independent variables presented a p-value < .20 at the bivariate analysis, multivariate linear regression was applied with subscale and total scale scores as the dependent variable. In this case, backward stepwise linear regression was performed ...”

I cannot follow this. Please rephrase. Why p<.20 exactly? multiple regression is done with backward stepwise regression. So that means that the aim is to determine the best subset of predictors in a set of predictors. What are the predictors? what is the dependent? Why focus on the best subset? Why not report results for all predictors as obtained in the regression analyses? What statistical criteria were used (F to remove etc.)?

“According to the multivariate linear regression, the following results are presented (Table 8):” It would be useful to state explicitly the full set of predictors. I would prefer to see the regression coefficients of the full set (rather than a subset).

Response 6:

Rephrase: “It is noted that for the selection of the independent variables entered in the multivariate logistic regression models, the value p<0.2 was used as a limit in order to limit to the minimum possible the possibility of not including in the models independent variables that were related to the dependent variables.”

Since we had a lot of independent variables (>15 and mainly sociodemographic characteristics), we started our analysis by examining bivariate correlations between those independent variables and the dependent variables of interest. In each case, we found that there were many meaningful correlations between our dependent variables and sociodemographic characteristics. The independent variables (13 to 15) were inserted in each multiple regression analysis (6 in total) and statistically significant findings are presented. Otherwise, tables containing all inserted variables would be quite extensive, without providing any useful information.

The criteria used in the regression analyses were the ones set as default by the statistical software program, that is, the levels of F’s probability, with a value of 0.05 for entry and a value of 0.10 for removal.

Point 7: The results concerning Cronbach’s alpha are reported in the stat method section. Don’t these belong in the results section? I never understand why Cronbach alphas are reported of well-developed psychometric instruments.

Response 7: Thank you for your comment. We appreciate your recommendation. Indeed, you are right that this tool is a reliable and valid widely used instrument. However, what we had in mind was that the value of Cronbach's alpha depends on the responses from a specific set of respondents, especially when they belong to a different cultural group than the original target population that the instrument was constructed for. This is why we thought appropriate to estimate the Cronbach for the specific population of the study.

The results concerning Cronbach’s alpha have been removed from the stat method section to the Results section.

Point 8: The regression results (Table 8) include the raw regression coefficients. It would be useful to include the standardized regression coefficients. Also, for each analyses, please include the R2 (proportion of explained variance). To understand the effect size, we require standardized betas and R2. The effect sizes (R2) are also important for the discussion of the implications of the results. If the R2 are low, then that has a bearing on the implications.

Response 8: R2 values have been added and standardized regression coefficients are now presented in table 8.

Point 10: The text would benefit from editing. Just a few examples:

“main reasons for being exercised” ... main reason for exercising.

“Respondents 87 have been classified into normal weight, overweight and obese individuals, according to 88 their BMI index.” ... were classified.

“Men exercised to a greater extent than women for championships”

Men exercised competitively to a greater extent than women

“Higher percentage of men compared to women reported being exercised for cham- 326 pionship reasons.”

Compared to women, more men reported to engage in exercise competitively. 

Response 10: corrected

Reviewer 2 Report

This is a clear manuscript assessing body image, BMI, social physique anxiety, and self-esteem among men and women involved in physical and athletic activities in Greece. The content is original, the statistical analysis is correct, and the sample size is large enough but it is a convenience sample.

The major problems that I have identified are 1- anthropometric characteristics which, while self-reported, need to be treated properly. First, the unit of measurement of stature is cm and not m. The authors should therefore correct the data if data were collected in cm, otherwise indicate it among the limitations of the study. Also, it does not make sense to report weight and stature averages in combined genders: Table 2 should be revised, indicating averages and SD separately by gender; 2- the discussion should be expanded by considering more recent literature on the topic (see, for example, doi: 10.1080/00223980.2013.846291  and doi: 10.3390/ijerph20065228 ); 3- I am not sure that the approvals of scientific boards of each athletic facility involved are sufficient -it will still be up to the journal to judge that.

Minor concerns:

-Line 71: remove the expression “and has been announced to be the European capital of  Culture for 2023” as it is not inherent.

-Table 2: replace N(%) with Mean (SD). Enter N(%) in the field next to " Have you ever had a problem with your body weight?”. Finally, replace kgr with kg.

-Lines 170-174: It would be preferable not to repeat in the text what is shown in the table but simply indicate the trends

Author Response

Response to Reviewer 2 Comments

Dear reviewer  

We would like to thank you for taking the necessary time and effort to review the manuscript. We sincerely appreciate all your valuable comments and suggestions, which helped us in improving the quality of the manuscript.

Point 1: anthropometric characteristics which, while self-reported, need to be treated properly. First, the unit of measurement of stature is cm and not m. The authors should therefore correct the data if data were collected in cm, otherwise indicate it among the limitations of the study.

Response 1: Although the unit of measurement of height is centimeters (cm) data have been selected in meters (m) for BMI calculation purposes (It is calculated by dividing a person's weight in kilograms by their height in meters squared). Referred among the limitations of the study.

Point 2: Also, it does not make sense to report weight and stature averages in combined genders: Table 2 should be revised, indicating averages and SD separately by gender;

Response 2: Table 2 was revised

Point 3: the discussion should be expanded by considering more recent literature on the topic (see, for example, doi: 10.1080/00223980.2013.846291 and doi: 10.3390/ijerph20065228); Response 3: The discussion has been expanded by considering the aforementioned more recent literature on the topic.

The reference Kong P, Harris LM. The sporting body: body image and eating disorder symptomatology among female athletes from leanness focused and nonleanness focused sports. J Psychol. 2015 Jan-Apr;149(1-2):141-60. doi: 10.1080/00223980.2013.846291. has been added in the discussion (line 329) and in the References section.

The reference Zaccagni, L.; Gualdi-Russo, E. The Impact of Sports Involvement on Body Image Perception and Ideals: A Systematic Review and Meta-Analysis. Int. J. Environ. Res. Public Health 2023, 20, 5228. https://doi.org/10.3390/ijerph20065228 has been added in the discussion (line 378) and in the References section.

Point 4: -Line 71: remove the expression “and has been announced to be the European capital of Culture for 2023” as it is not inherent.

Response 4: corrected

Point 5: -Table 2: replace N (%) with Mean (SD). Enter N (%) in the field next to " Have you ever had a problem with your body weight?”. Finally, replace kgr with kg.

Response 5: corrected

Point 6: Lines 170-174: It would be preferable not to repeat in the text what is shown in the table but simply indicate the trends

Response 6: corrected

Reviewer 3 Report

Diseases-2335792 Reviewer Comments

Title: Body Image, Social Physique Anxiety levels and Self-Esteem among Adults Participating in Physical Activity Programs

Reviewer Comments:

Summary of Manuscript:

This cross-sectional study aimed to investigate body image, body mass index (BMI), social physique anxiety, self-esteem, and possible correlations between the above factors. The sample included 245 adults who participated in gym trainings and athletic activities in an area of Greece. Participants completed: 1) a sociodemographic questionnaire; 2) BMI values; 3) the Body-Esteem Scale for adolescents and adults; 4) the Social Physique Anxiety Scale; and 5) the Rosenberg Self-Esteem Scale. Results indicated that females and individuals with higher BMI reported a statistically significant lower body esteem and greater social physique anxiety levels compared to males and individuals with lower BMI, respectively. Significant differences were noted between body esteem and 1) social physique anxiety levels; 2) age; 3) BMI values; and 4) self-reports of never having been “overweight” or “underweight” according to the BMI scale. Additionally, individuals with lower body esteem and greater social physique anxiety levels presented with lower global self-esteem. Limitations include: 1) sample located in a single geographical region of Greece (affecting generalizability of results); 2) use of self-reports (potential bias towards socially desirable responses); 3) cross-sectional study design (precludes causal associations); and 4) lack of a control group of non-exercisers for comparisons. The authors concluded that the findings of the study could be used by professionals (health and athletic) to provide a deeper understanding of the relationship between body image and exercise. This could foster empathy, as well as lead to the implementation of strategies that decrease body dissatisfaction and increase self-esteem, which in turn could reduce the future risk for the development of eating and mood disorders.

Overall, the manuscript is worthy of publication, and will make a valuable contribution to the literature by increasing our knowledge about the association between body image, BMI, social physique anxiety, and self-esteem. However, there are some minor areas that could be improved (e.g., language, format) before the manuscript is accepted. Please see below for more detailed comments. In many cases, the comments apply throughout the manuscript, so once you make the first correction, you can use it going forwards. I believe these revisions will further strengthen the manuscript.

Language:  

*Abstract/Text/Tables:

*Please consistently use the more person-centered phrase of “participants” (vs. “subjects”) throughout the manuscript.

*Please temper your strong/conclusive language when you are expressing your own opinions. For example:

“Individuals need to be involved in physical and athletic activities…” (please revise to “individuals may benefit from being involved in physical and athletic activities…”)

“A great number…become socially accepted.” (please revise to “increase their perception of social acceptance.”)

“…in order to achieve optimal outcomes and ensure individuals’ best quality of life.” (please revise to “in order to improve outcomes and promote a higher quality of life.”)

*Text:

*I would like to acknowledge that the authors are from Greece, and they have done an excellent job overall with translating the manuscript to English. That being said, there are some minor grammatical errors and colloquial phrases that could be revised for a wider (predominantly native English) readership. For example:

Grammatical errors:

“The scale has been used in Greek population.” (please revise to “used in the Greek population”)

“In addition, when individuals evaluate more positively the way that others perceive their appearance, have lower social physique anxiety and higher self-esteem.” (please revise to “In addition, when individuals believe that others perceive their appearance to be more attractive, they tend to have lower social physique anxiety and higher self-esteem.”)

Colloquial phrases:

“being exercised” (please revise to “exercising”)

“the study/scale of (author, et al., year)” (please revise to “the study scale by (author, et al. year)”

“championships” (please revise to “athletic competitions”)

Format:

*Please spell out all acronyms at first usage in the Abstract/Highlights/Text, and then use these acronyms consistently afterwards (vs. switching back to spelling them out). For example:

-Body Mass Index (BMI)

-Standard Deviation (SD)

*Please break up run-on sentences that are difficult for the reader to follow. For example:

Lines 37-42: “It has been found that…physical activity.”

Lines 145-150: “All subjects…Association.”

Lines 269-274: “A large percentage…concerned.”

*Please bold font the Table citations in the text for ease in readability.

*All Tables should be able to stand alone from the text; therefore, the titles should be clear (more descriptive), and all acronyms should be spelled out in the footnotes.

*Please provide more information for the reader in these areas:

*Please reaffirm that the questionnaire was self-report. Even though it includes some validated measures, there was no corresponding clinical interview, which may limit the reliability.

*Correspondingly, please also reaffirm that the study had a cross-sectional design, so causal associations cannot be drawn from the findings.

*Please clarify the origin of BMI, and how the ranges are labeled as “underweight,” “normal weight,” “overweight,” and “obese” (add citations). In the USA in particular, the use of these terms has become sensitive for many reasons, and it is important to note that you are using these terms based upon a widely-used (yet controversial) metric. Relatedly, I would also suggest rephrasing “Have you ever had a problem with your body weight?” to “Have you ever been classified as “underweight,” “overweight,” or “obese” according to the BMI index?” Though the study has already been completed, if that was the metric that you conveyed to the participants, please specify that here (otherwise, was it up to their subjective determination of “problem”?).

*Please clarify why you specify in the abstract conclusions that “engagement in physical activity promotes physical as well as mental well-being, contributing to improved quality of life, which may the most important issue for nursing science…” Are you writing to nursing science readers?

*The introduction/discussion sections on the connections between physical activity, mood, and neurobiology of the brain (neurotransmitters and areas) are very enlightening. I would highlight these more in the manuscript (e.g., add them to the conclusions, propose related future research directions for the reader, etc.). 

*Please also elaborate upon the social representations of women vs. men, as well as differences in gonadal hormones, and how this may contribute to the relative prevalence of anxiety.

*Please clarify that the majority of your participants engaged in athletic activities had their professional career in these sports, compared to the majority of participants engaged in gym training were not otherwise exercising. This affects the results that you are presenting.

*Please define at first use the definitions of all of the terms used in the manuscript. For example:

Self-esteem

Social physique anxiety

Body image

*Please delete erroneous information from the manuscript. For example:

“Municipality of Elefsina…has been announced to be the European capital of Culture for 2023.”

I hope that these comments are helpful, and will assist with further strengthening this manuscript.

Author Response

Response to Reviewer 3 Comments

Dear reviewer  

We would like to thank you for taking the necessary time and effort to review the manuscript. We sincerely appreciate all your valuable comments and suggestions, which helped us in improving the quality of the manuscript.

Point 1: Please consistently use the more person-centered phrase of “participants” (vs. “subjects”) throughout the manuscript.

Response 1: corrected

Point 2: Please temper your strong/conclusive language when you are expressing your own opinions. For example:

“Individuals need to be involved in physical and athletic activities…” (please revise to “individuals may benefit from being involved in physical and athletic activities…”)

“A great number…become socially accepted.” (please revise to “increase their perception of social acceptance.”)

“…in order to achieve optimal outcomes and ensure individuals’ best quality of life.” (please revise to “in order to improve outcomes and promote a higher quality of life.”)

Response 2: corrected

Point 3: “The scale has been used in Greek population.” (please revise to “used in the Greek population”)

Response 3: corrected

Point 4: “In addition, when individuals evaluate more positively the way that others perceive their appearance, have lower social physique anxiety and higher self-esteem.” (please revise to “In addition, when individuals believe that others perceive their appearance to be more attractive, they tend to have lower social physique anxiety and higher self-esteem.”)

Response 4: corrected

Point 5: “being exercised” (please revise to “exercising”)

Response 5: corrected throughout the manuscript.

Point 6: “the study/scale of (author, et al., year)” (please revise to “the study scale by (author, et al. year)”

Response 6: corrected throughout the manuscript.

Point 7: “championships” (please revise to “athletic competitions”)

Response 7: corrected throughout the manuscript.

Point 8: *Please spell out all acronyms at first usage in the Abstract/Highlights/Text, and then use these acronyms consistently afterwards (vs. switching back to spelling them out). For example:

-Body Mass Index (BMI)

-Standard Deviation (SD)

Response 8: corrected throughout the manuscript.

Point 9: *Please break up run-on sentences that are difficult for the reader to follow. For example:

Lines 37-42: “It has been found that…physical activity.”

Lines 145-150: “All subjects…Association.”

Lines 269-274: “A large percentage…concerned.”

Response 9: corrected

Point 10: *Please bold font the Table citations in the text for ease in readability.

Response 10: corrected

Point 11: *All Tables should be able to stand alone from the text; therefore, the titles should be clear (more descriptive), and all acronyms should be spelled out in the footnotes.

Response 11: corrected

Point 12: *Please reaffirm that the questionnaire was self-report. Even though it includes some validated measures, there was no corresponding clinical interview, which may limit the reliability.

Response 12: corrected (line 93 in the section of Instruments and 481-483 in Limitations)

Point 13: *Correspondingly, please also reaffirm that the study had a cross-sectional design, so causal associations cannot be drawn from the findings.

Response 13: corrected (lines 484-485 in Limitations)

Point 14: *Please clarify the origin of BMI, and how the ranges are labeled as “underweight,” “normal weight,” “overweight,” and “obese” (add citations). In the USA in particular, the use of these terms has become sensitive for many reasons, and it is important to note that you are using these terms based upon a widely-used (yet controversial) metric.

Response 14: The following sentences have been added in Instruments (line 99- ) and the citation in References.

The BMI ranges are categorized as follows: (a) Underweight: BMI less than 18.5; (b) Normal weight: BMI between 18.5 and 25; (c) Overweight: BMI between 25 and 30; (d) Obese: BMI 30 or greater. It is important to note that while BMI is a widely-used and useful screening tool for assessing weight-related health risks, it has its limitations and should not be the sole determinant of an individual's health status [14].

[14] Centers for Disease Control and Prevention. About Adult BMI. (2021). Retrieved from https://www.cdc.gov/healthyweight/assessing/bmi/adult_bmi/index.html

Point 15: Relatedly, I would also suggest rephrasing “Have you ever had a problem with your body weight?” to “Have you ever been classified as “underweight,” “overweight,” or “obese” according to the BMI index?” Though the study has already been completed, if that was the metric that you conveyed to the participants, please specify that here (otherwise, was it up to their subjective determination of “problem”?).

Response 15: it was up to their subjective determination of “problem”

Point 16: *Please clarify why you specify in the abstract conclusions that “engagement in physical activity promotes physical as well as mental well-being, contributing to improved quality of life, which may the most important issue for nursing science…” Are you writing to nursing science readers?

Response 16: corrected. We are not writing to nursing science readers.

Point 17: *The introduction/discussion sections on the connections between physical activity, mood, and neurobiology of the brain (neurotransmitters and areas) are very enlightening. I would highlight these more in the manuscript (e.g., add them to the conclusions, propose related future research directions for the reader, etc.). 

Response 17: We added a section on the connections between physical activity, mood, and neurobiology of the brain to the conclusions. Related future research directions have been proposed.

Point 18: *Please also elaborate upon the social representations of women vs. men, as well as differences in gonadal hormones, and how this may contribute to the relative prevalence of anxiety.

Response 18: added in conclusion

Point 19: *Please clarify that the majority of your participants engaged in athletic activities had their professional career in these sports, compared to the majority of participants engaged in gym training were not otherwise exercising. This affects the results that you are presenting.

Response 19: clarified in the text

Point 20: *Please define at first use the definitions of all of the terms used in the manuscript. For example:

Self-esteem

Social physique anxiety

Body image

Response 20: definition of self- esteem (line 49-51), definition of Body image (line 52-55), definition of Social physique anxiety (line 56-58). 

Point 21: *Please delete erroneous information from the manuscript. For example:

“Municipality of Elefsina…has been announced to be the European capital of Culture for 2023.”

Response 21: corrected

Round 2

Reviewer 1 Report

Thanks for considering and acting on my comments. Here are my residual comments. 

"The group of individuals recruited at gyms in the above study may be of interest for several reasons. Firstly, individuals who attend gyms and sports schools are likely to be more physically active than the general population, which makes them an important target group for interventions to promote positive body image and engagement in physical activity. "

The people who go to gyms are an important target group ... to promote engagement in physical activity". I do not understand this, because the people to attend gyms DO engage in physical activity. The important target group would seem to be people who DO NOT engage in physical activity. 

"There was no respective technique used so as to maintain a desired error rate while preserving the overall alpha level. Nevertheless, while the choice of 0.05 as the significance threshold is somewhat arbitrary, the rationale for using an alpha level of 0.05 was to balance Type I and Type II errors as a reasonable choice and due to practicality and ease of interpretation; it is simple to understand and apply and provides a reasonable trade-off between the need for evidence and the resources required to obtain that evidence."

The statement that the .05 (uncorrected for multiple testing) provides "a balance between Type I and Type II errors" suggests that you have evaluated the Type I and Type II error rates in this study. So please 1) state explicitly in the paper your policy w.r.t alpha and include information concerning you evaluation of type I and type II rates in this study. The statement .05 is " simple to understand and apply" is hard to follow. It seems to imply that  alpha=.05 is easy to understand and apply, but alpha = .01 (corrected for 5 tests) is not easy to understand and apply. How does that work?   

Author Response

Response to Reviewer 1 Comments

Dear reviewer,

once again, we would like to thank you for taking the necessary time and effort to consider our previous answer. We do hope that you will find our rationale to be acceptable.

Point 1: "The group of individuals recruited at gyms in the above study may be of interest for several reasons. Firstly, individuals who attend gyms and sports schools are likely to be more physically active than the general population, which makes them an important target group for interventions to promote positive body image and engagement in physical activity."

The people who go to gyms are an important target group ... to promote engagement in physical activity". I do not understand this, because the people to attend gyms DO engage in physical activity. The important target group would seem to be people who DO NOT engage in physical activity.

Response 1: Individuals who attend gyms and sports schools are likely to be more physically active than the general population. This group is significant in terms of comprehending their reasons for exercising, as this knowledge could be used to develop effective interventions aimed at encouraging those who are not currently physically active to engage in physical activity.

The group of individuals who attend gyms and sports schools may also be of interest in order to explore the impact of physical activity and different types of exercise on aspects such as mood, self-esteem, and body image. By examining this group, we can gain a better understanding of how exercise affects these factors, which can help in developing effective interventions to promote sustained participation in physical activity.

Point 2: "There was no respective technique used so as to maintain a desired error rate while preserving the overall alpha level. Nevertheless, while the choice of 0.05 as the significance threshold is somewhat arbitrary, the rationale for using an alpha level of 0.05 was to balance Type I and Type II errors as a reasonable choice and due to practicality and ease of interpretation; it is simple to understand and apply and provides a reasonable trade-off between the need for evidence and the resources required to obtain that evidence."

The statement that the .05 (uncorrected for multiple testing) provides "a balance between Type I and Type II errors" suggests that you have evaluated the Type I and Type II error rates in this study. So please 1) state explicitly in the paper your policy w.r.t alpha and include information concerning you evaluation of type I and type II rates in this study. The statement .05 is " simple to understand and apply" is hard to follow. It seems to imply that alpha=.05 is easy to understand and apply, but alpha = .01 (corrected for 5 tests) is not easy to understand and apply. How does that work?  

Response 2:

We would like to highlight the fact that our study was purely exploratory, as stated by its aim (Abstract and Introduction: The aim of this study was to investigate body image, BMI characteristics, social physique anxiety, self-esteem and possible correlations between the above factors in individuals participating in exercise programs and athletic activities). There were no endpoint(s) to confirm. Therefore, we had no hypotheses to reject or fail to reject and our analyses were not guided by any such hypotheses. That means that there weren’t any predefined set (or family) of tests that needed to be performed and leading to a certain end.

In the light of the exploratory nature of the present study, the authors have a priori decided to present all findings arising from various individual statistical tests while maintaining a statistically significant level of p<0.05. However, it could be argued that there should be a provision for inflated type I error (false positive findings).

On the other hand, as mentioned earlier, our aim was to unearth any correlations between the concepts under examination, without having set any endpoint(s). Therefore, by applying correction technics to avoid type I error, critical (in our opinion) findings would have been left out from our study. It was our intention to highlight every possible contributing factor for the interested researchers in order to take them into account and formulate a specific hypothesis (-es) to verify it (them).

The above could be based on the literature (for example, Bender & Lange. (2001). Adjusting for multiple testing—when and how? Journal of Clinical Epidemiology, 54, 343–349 and Schochet, P.Z. (2008) Technical Methods Report: Guidelines for Multiple Testing in Impact Evaluations. Available at: https://ies.ed.gov/ncee/pubs/20084018/chapter_2.asp), although, there aren’t any universally accepted conclusive guidelines regarding exploratory studies.

Accordingly, we have enhanced our limitations section in order to inform interested readers.

Reviewer 2 Report

The authors revised the manuscript in accordance with the suggestions provided. The manuscript is ready for publication.

Author Response

Dear reviewer,  

once again, we would like to thank you for taking the necessary time and effort to review the manuscript. We sincerely appreciate all your valuable comments and suggestions, which helped us in improving the quality of the manuscript.